# Quantitative Multi-Parametric MRI of the Prostate Reveals Racial Differences

**DOI:** 10.3390/cancers16203499

**Published:** 2024-10-16

**Authors:** Aritrick Chatterjee, Xiaobing Fan, Jessica Slear, Gregory Asare, Ambereen N. Yousuf, Milica Medved, Tatjana Antic, Scott Eggener, Gregory S. Karczmar, Aytekin Oto

**Affiliations:** 1Department of Radiology, University of Chicago, Chicago, IL 60637, USA; xfan@uchicago.edu (X.F.); ayousuf@uchicagomedicine.org (A.N.Y.); mmedved@uchicago.edu (M.M.); gskarczm@uchicago.edu (G.S.K.); aoto@bsd.uchicago.edu (A.O.); 2Sanford J. Grossman Center of Excellence in Prostate Imaging and Image Guided Therapy, University of Chicago, Chicago, IL 60637, USA; 3Department of Pathology, University of Chicago, Chicago, IL 60637, USA; tatjana.antic@bsd.uchicago.edu; 4Section of Urology, University of Chicago, Chicago, IL 60637, USA; seggener@bsd.uchicago.edu

**Keywords:** prostate cancer, MRI, racial differences, quantitative, DCE-MRI, African Americans, Caucasian Americans

## Abstract

This study investigated whether quantitative MRI and histology of the prostate reveal differences between races that can affect diagnosis. The cancer signal enhancement rate (α) on dynamic contrast-enhanced MRI (DCE-MRI) was significantly higher for African Americans (AAs) compared to Caucasian Americans (CAs). The signal washout rate (β) was significantly lower in benign tissue in AAs and significantly elevated in cancers in AAs. However, no significant differences were found for ADC and T2. The ROC analysis showed that the apparent diffusion coefficient (ADC) and T2 are slightly less effective in AAs compared to CAs. DCE significantly improves the differentiation of PCa from benign in AAs (α: 52%, β: 62% more effective in AAs compared to CAs). Histologic analysis showed that cancers have a greater proportion of epithelium and lower lumen in AAs compared to CAs. This study shows that the different races have different quantitative MRI values and histologic makeup. Quantitative DCE-MRI is highly effective and improves PCa diagnosis in African Americans.

## 1. Introduction

Prostate cancer (PCa) is the second most commonly diagnosed cancer in men, with 1 in 8 men diagnosed with it, and is one of the leading causes of cancer-related deaths worldwide [1,2]. PCa incidence and mortality are strongly associated with age, with an average age of diagnosis at 67 years old [3]. In addition to age, there are inherited and genetic risk factors for prostate cancer. Men with a family history of prostate cancer (inherited risk factors), genetic risk factors (germline mutations—BRCA2 gene), and specific race (African ancestry) have an elevated risk for developing prostate cancer [3,4,5,6]. 

Race is an important factor in prostate cancer risk and incidence. African Americans (AAs) have a higher risk and incidence for prostate cancer compared to Caucasian Americans (CAs), while PCa occurs less often in Asian Americans, Hispanics, and Latino men than in non-Hispanic Caucasian men. Specifically for AAs, this higher risk for PCa is likely due to a combination of factors, including underlying tumor differences (aggressiveness), genetic factors, and racial disparities in access to healthcare [5,7,8,9]. This is a critical problem, especially because PCa detected in AAs is typically more aggressive and advanced compared to PCa in CAs, and the mortality rate for AAs with PCa compared to CAs is significantly higher, even after adjusting for prognostic factors (PSA level, age, clinical stage, and Gleason score) [10,11]. A recent study found that racial disparities were greatest in non-clinically significant low-grade Gleason 6 disease, in which AA men were twice as likely to die of Gleason 6 tumors compared to men from other ethnicities [5]. There are no studies that have clearly established racial differences in prostate cancer biology and physiology that could affect diagnosis using imaging methods.

MRI can play an important role in the diagnosis and management of PCa because it is non-invasive, can reliably detect clinically significant tumors, and can provide information regarding tumor size, location, and grade [12]. The current diagnostic radiology guideline for prostate MRI is the Prostate Imaging—Reporting and Data System (PI-RADS v2.1) [13]. This system utilizes T2-weighted (T2W) imaging and diffusion-weighting imaging (DWI) as the primary sequences, while there is less emphasis on dynamic contrast-enhanced (DCE) MRI, which is used as a secondary sequence. It is also important to note that the PI-RADS guidelines were recommended by a steering committee that is almost entirely comprised of experts from the US and Europe, and most of the published prostate MRI research has been carried out predominantly on Caucasian populations [14]. While the PI-RADS guidelines may provide accurate diagnoses for certain populations (Caucasians), we should not assume that the same solution will work for other patient populations as well, such as African Americans who are at a higher risk for PCa [15] and Asians who are at a lower risk for PCa [16]. This is especially important, as physician decision bias and disparities in access to healthcare exist for AA patients. AAs are less likely to undergo MRI imaging following an elevated prostate-specific antigen (PSA) compared to CAs [17]. 

A recent study by Mahran et al. [18] showed a racial disparity in the utility of qualitative multi-parametric MRI for the diagnosis of PCa. It is likely that the poor performance in PCa diagnosis is due, in part, to the failure to account for racial differences in prostate and PCa biology and physiology. However, no studies have evaluated the racial differences in biomarkers/parameters from quantitative MRI. Whether PCas have different MR characteristics in AA versus CA men has yet to be determined. MRI-based quantitative biomarkers that account for racial differences could be used to create MRI protocols specifically for different racial groups. Therefore, this study investigates whether quantitative MRI and quantitative histology of the prostate reveal differences between races, specifically African Americans and Caucasian Americans, that can affect diagnosis.

## 2. Materials and Methods

***Study participants:*** This study involved retrospective analysis of prospectively acquired data. This study was approved by the institutional review board and conducted with informed patient consent, and was HIPPA compliant. Inclusion criteria for the study included patients with known or suspected PCa who underwent prostate mpMRI between February 2014 and November 2021 at the MRI Research Center prior to undergoing prostatectomy or biopsy. Patients were excluded from the study if they had undergone radiation, chemotherapy, or hormonal therapy, as these therapies can alter the MRI signal. Patients self-reported their race. For further analysis, we chose the two races in this cohort with the largest sample size: Caucasian Americans and African Americans. 

***MRI Acquisition:*** Patients underwent a preoperative mpMRI scan on the 3T Philips Ingenia or Achieva MR scanners with the use of a 16-channel phased array coil around their pelvis (Philips, Eindhoven, The Netherlands) and an endorectal coil (Medrad, Warrendale, PA, USA). The MR protocol included T2-weighted images (axial, coronal) and axial multi-echo T2-weighted, diffusion-weighted, and dynamic contrast-enhanced images (DCE). For DCE MR imaging, either gadobenate dimeglumine (Multihance, Bracco, Milan, Italy) or gadoterate meglumine (Dotarem, Guerbet, Paris, France) of 0.1 mmol/kg was injected, followed by a 20 mL saline flush. Since the quantitative MR parameters depend on imaging parameters (b-value, echo time, etc.) [19,20,21], we chose a cohort where all patients underwent a very similar MRI protocol so that the quantitative parameters could be compared. Typical imaging parameters for the cohort used in this study are described in Table 1. 

***Reference standard:*** Post imaging, patients with known cancer underwent radical prostatectomy, and patients with suspected cancer underwent 12 core systematic biopsies along with MR-TRUS biopsies based on the assessment of radiologists using the PI-RADS guidelines. The prostatectomy samples were fixed overnight in 10% buffered formalin, transversally sectioned every 4 mm, and then used to create whole-mount histology slides. The biopsy samples from patients with suspected PCa were also processed to create histological slides. All slides were H&E stained and evaluated for cancer by an expert pathologist (TA, 18 years experience). Cancers were outlined, and the Gleason score for each lesion was assigned. The prostatectomy slides were also digitized at 20× magnification using a brightfield Olympus VS120 whole-mount digital microscope (Olympus, Waltham, MA, USA), which was to be used for the subsequent quantitative histologic analyses described later.

***MRI analysis:*** Histology slides and MR images were correlated with the consensus of an expert radiologist, a pathologist, and a medical physicist (AO, TA, and AC with 20, 18, and 9 years of experience, respectively). Only cancers larger than the threshold 5 mm × 5 mm on the prostatectomy specimens and 5 mm on mpMRI for the biopsy cases were used for analysis. Using a custom PCampReviewer module on a 3D slicer [22] and T2-weighted images as a reference, the apparent diffusion coefficient (ADC) and DCE MR images were co-registered and matched with the corresponding histological portions. The axial T2-weighted images were marked by a radiologist (AO, 20 years experience) for sections with a pathologist-verified PCa and benign tissue. The radiologist used the pathology whole-mount histology cancer outlines as a guide to drawing ROIs on the T2-weighted MR images. These ROIs were then transferred to the other mpMRI sequences using 3D Slicer to maintain the same PCa shape and size.

Quantitative analysis of the mpMRI data was performed in MATLAB (MathWorks, Natick, MA, USA) using in-house programs. This code calculated the ADC and quantitative T2 (T2 relaxation or spin–spin relaxation time) maps using a mono-exponential signal decay model on a voxel-by-voxel basis using the diffusion-weighted and multi-echo T2-weighted images. Quantitative analysis of the DCE-MRI data was conducted using the empirical mathematical model (EMM) as described in Fan et al. [23]. The EMM makes no assumptions about the underlying physiology of a tumor and provides quantitative analysis of the kinetics that the radiologist sees on visual assessment. The percent signal enhancement (*PSE*) curve was calculated using the formula below where *S*_0_ = the baseline signal intensity and *S*(*t*) = the DCE signal at time ‘*t*’.
$$PSE\left(t\right)=\frac{S\left(t\right)-S_{0}}{S_{0}}\times 100$$

Then, *PSE* vs. the time curves were analyzed using the following equation
$$PSE\left(t\right)=A\left(1-e^{-\alpha t}\right)e^{-\beta t}$$
to estimate the amplitude of the relative percent signal enhancement or *PSE* curve or *A*, the signal enhancement rate or *α*, and the signal washout rate or *β*.

***Quantitative histology:*** We used the digitized H&E stained prostatectomy samples for subsequent quantitative histologic analysis, similar to previous work that validated this approach [24,25]. ROIs were selected using ImageJ (National Institutes of Health, Bethesda, MD, USA) for the confirmed cancers and benign tissue in the peripheral and transition zones corresponding to the ROIs taken for quantitative MRI analysis earlier. Using the “Smart Segment” functionality in Image Pro (Media Cybernetics, Rockville, MD, USA), the images were segmented into the three constituents that make up prostate tissue: stroma, epithelium, and lumen on the basis of color, intensity, morphology, and background. The results were iteratively rectified semi-automatically by the consensus of a pathologist and medical physicist until the final segmented image was determined to have an error of less than 5%. Then, using the “Count” functionality, the percentage volumes of these prostatic tissue components were calculated.

***Statistical analysis:*** Statistical analysis was performed using SPSS v29 (IBM Corporation, Armonk, NY, USA). The difference in measured mpMRI and quantitative histoloygy parameters between AAs and CAs was assessed by *t*-test. The chi-squared test was used to test if the distribution was different between the two cohorts. Receiver operating characteristic (ROC) analysis was used to evaluate the performance of the quantitative mpMRI parameters in diagnosing PCa. The area under the ROC curve (AUC) and ideal cutoff point or the Youdens index with associated sensitivity and specificity were reported. The Z-score test was used to determine the significant difference in the AUC value between the two populations.

## 3. Results

***Patient and lesion characteristics:*** The final cohort for this study included 47 African American and 98 Caucasian American subjects. No significant difference in age (*p* = 0.29) and PSA level (*p* = 0.94) was found between the AAs (age = 60 ± 7 years, PSA = 8.5 ± 5.9 ng/mL) versus CAs (age = 58 ± 8 years, PSA = 8.6 ± 9.9 ng/mL). The percentage of patients undergoing biopsy and prostatectomy were similar (*p* = 0.89) between the two groups, with 25 biopsies (53%) and 22 prostatectomy cases (47%) for AAs and 51 (52%) and 47 (48%), respectively, in CA patients. AAs (29% Gleason 3 + 3, 36% Gleason 3 + 4, 22% Gleason 4 + 3, 11% Gleason 4 + 4, 2% Gleason 4 + 5) had a greater (*p* = 0.01) percentage of higher Gleason-grade lesions compared to CAs (29% Gleason 3 + 3, 57% Gleason 3 + 4, 11% Gleason 4 + 3, 1% Gleason 4 + 4, 2% Gleason 4 + 5). Detailed patient and lesion characteristics can be found in Table 2. 

***Quantitative mpMRI results:*** Representative figures depicting prostate MRI for a typical AA and CA patient in this study are shown in Figure 1 and Figure 2, respectively. We found no significant difference in the quantitative ADC values (Cancers: 1.03 ± 0.32 µm^2^/ms in AA vs. 1.07 ± 0.34 µm^2^/ms in CA, *p* = 0.75, Benign: 1.53 ± 0.37 in AA vs. 1.62 ± 0.37 µm^2^/ms in CA, *p* = 0.12) and T2 values (Cancer: 107.5 ± 55.8 ms in AA vs. 99.7 ± 27.7 ms, in CA *p* = 0.25, Benign: 151.9 ± 96.5 ms for AA, 159.1 ± 73.2 ms for CA, *p* = 0.70) between AAs and CAs for cancer and benign tissue. However, the ADC and T2 values for cancers were nominally higher in AAs than in CAs despite the higher Gleason-grade cancers in the AA cohort. No significant difference was found between AAs and CAs for these metrics when compared for each Gleason score category. The detailed results for quantitative MRI metrics measured in this study can be found in Table 3. 

However, significant differences were found in the quantitative DCE metrics between the two cohorts. The DCE signal enhancement rate (α) was significantly higher in cancerous tissue for AAs compared to CAs (AA: 13.3 ± 9.3 s^−1^ vs. CA: 6.1 ± 4.7 s^−1^, *p* < 0.001). Similarly, differences were found across all Gleason scores (Gleason 3 + 3: 10.3 ± 4.4 vs. 4.9 ± 2.9 s^−1^, *p* = 0.002; Gleason 3 + 4: 11.7 ± 6.5 vs. 6.7 ± 5.6 s^−1^, *p* = 0.04; and Gleason ≥ 4 + 3: 15.6 ± 11.7 and 6.2 ± 2.9 s^−1^, *p* = 0.02, respectively) where the signal enhancement rate was significantly higher in AAs than CAs. No significant differences in the signal enhancement rate were found in benign tissue (5.1 ± 4.6 for AA vs. 4.9 ± 2.9 s^−1^ for CA, *p* = 0.81). 

The DCE signal washout rate (β) was significantly lower in the benign tissue of AAs (AA: 0.01 ± 0.09 s^−1^ vs. CA: 0.07 ± 0.07 s^−1^, *p* < 0.001) and was significantly elevated in cancer tissue in AAs (AA: 0.12 ± 0.07 s^−1^ vs. CA: 0.07 ± 0.08 s^−1^, *p* = 0.02). However, while these differences were found to be significant in cancer ROIs overall for AAs versus CAs, no significant differences were found across the Gleason score categories. 

Significant differences in the DCE signal enhancement amplitude (A) were found between different Gleason grades between AAs and CAs, except for Gleason 3 + 4 tumors. In the cancer ROIs overall, the AA signal enhancement amplitude (181.1 ± 44.8%) was significantly higher (*p* < 0.001) compared to CAs (120.2 ± 47.4%). In benign tissue, the amplitude was also significantly higher (*p* < 0.001) for AAs (157.6 ± 82.6%) compared to CAs (126.1 ± 51.2%) for CAs. 

***Diagnostic performance:*** The diagnostic performance, as evidenced by the area under the ROC curve, showed that ADC (AA = 0.79 vs. CA = 0.87, *p* = 0.21) and T2 (AA = 0.68 vs. CA = 0.79, *p* = 0.15) were statistically equally effective in differentiation between the benign and cancer tissues in both CAs and AAs. However, it should be noted that the ADC (AUC 10% lower in AAs) and T2 (AUC 16% lower in AAs) were nominally less effective in AAs compared to CAs for the diagnosis of PCa in CAs despite a higher proportion of high-grade cancer in the AA cohort. DCE, on the other hand, significantly improved the differentiation of PCa from benign in AAs but was found to be ineffective in CAs. The area under the ROC curve showed that the DCE signal enhancement rate (AUC for AAs = 0.88 vs. CAs = 0.58, *p* < 0.001) and signal washout rate (AUC for AAs = 0.81 vs. CAs = 0.50, *p* < 0.001) were 52% and 62% significantly more effective (*p* < 0.001), respectively, in diagnosing PCa in AA patients. In addition, the cutoff values based on the Youdens index for the quantitative mpMRI parameters were different for the two cohorts. This leads to a different sensitivity and specificity measured using these cutoffs (the detailed results are in Table 4).

***Quantitative histology:*** The histology analysis for prostate tissue composition showed a similar breakdown of tissue components between AAs (epithelium 28.7 ± 9.0%, lumen 28.8 ± 13.3%, stroma 42.3 ± 10.2%) and CAs (epithelium 29.6 ± 9.2%, lumen 27.4 ± 11.1%, stroma 43.1 ± 12.1%) for benign tissue. However, in cancerous tissue, there were greater proportions of epithelium and lower lumen (*p* = 0.04) in CAs (epithelium 50.9 ± 12.3%, lumen 10.5 ± 6.9%) compared to AAs (epithelium 44.7 ± 12.8%, lumen 16.2 ± 6.8%), suggesting differences in the histologic makeup and micro-anatomy of PCa in AAs versus CAs. No difference in stroma volume was found for cancer between the two groups. The detailed results are in Table 5.

## 4. Discussion

The results of our study show that the cancer signal enhancement rate (α) of prostate cancer is significantly higher for AAs compared to CAs. The DCE signal washout rate (β) is significantly lower in the benign tissue of AAs and significantly elevated in the cancer tissue of AAs. Due to these differences, DCE significantly improves the differentiation of PCa from benign prostate tissue in AAs but not in CAs. There were no significant differences in the quantitative ADC and T2 values between AAs and CAs. The histologic analysis showed that cancers have a greater proportion of epithelium and lower lumen in CAs compared to AAs. These findings underscore the importance of considering racial differences when developing screening or diagnostic guidelines, especially as bi-parametric MRI is increasingly being proposed for population screening.

PCa detected in AAs tends to have higher Gleason score lesions (more aggressive and advanced, Gleason 4 + 3 and above) compared to sPCa in CAs [10,11]. We found a similar trend in our cohort. Despite having more high-grade lesions in AA, cancers in AAs tend to have very similar ADC and T2 values to CAs. The background benign tissue has a nominally lower T2 and ADC in AAs. This potentially decreases the contrast between the cancer and normal regions in the prostate and can make the cancer less conspicuous. [26] This is evidenced by the lower AUC values for cancer detection using ADC and T2 in AAs. This is consistent with the qualitative mpMRI results from Mahran et al., where the negative predictive value for AAs using mpMRI is lower than that for CAs [18]. Our quantitative histology results showed that cancer in AAs tend to be less dense cancers that have lower epithelium (epithelium is associated with lower ADC and T2 values) and higher lumen (lumen is associated with higher ADC and T2 values), making them less conspicuous on T2 and ADC. In addition, increased inflammation in the tumor microenvironment of prostate cancer in AA men has been noted; this is a driver of disparate clinical outcomes [27]. From the MRI literature, we also know that inflammation affects T2 relaxometry and diffusion measures and can mimic prostate cancer, making cancer diagnosis more difficult [28].

Quantitative DCE-MRI using EMM has been shown to be effective in the diagnosis of PCa [29,30]. The results of this study demonstrate that EMM is even more effective for differentiating PCa from benign tissue in AAs, with the AUCs for α and β being 52% and 62% greater, respectively, in AAs compared to CAs. Numerous studies have noted inherent molecular and biological differences in cancer of AA patients [31,32]. The observed difference in contrast uptake and washout can be attributed to the tumor microenvironment. Most importantly, neo-angiogenesis or higher microvessel density (MVD) in cancer of AA subjects compared to CAs has been reported [32]. These blood vessels, produced by cancers, allow an increased blood flow, which results in rapid contrast uptake and quick washout. The increased blood flow brings increased oxygen and nutrients that support tumor growth, invasion, and metastatic progression. This is consistent with findings of increased tumor progression and worse clinical outcomes in AAs [33]. 

This study found that the ideal cutoff values (Youdens index) for the quantitative mpMRI parameters were very different for AAs vs. CAs, showing that the standard diagnostic model is not optimal for AAs. This also suggests the more general hypothesis that quantitative MRI protocols and thresholds for PCa diagnosis for different races should be determined independently. The current PI-RADS v2.1 guideline [13] utilizes T2-weighted (T2W) imaging and diffusion-weighting imaging (DWI) as the primary sequences while giving less emphasis to dynamic contrast-enhanced (DCE) MRI. The results showing that DCE-MRI is highly effective for PCa diagnosis in African Americans suggest that the PI-RADS guidelines should be modified to include a greater emphasis on DCE-MRI for AAs. However, the current results must be verified in a larger cohort in a multi-center setting.

There is growing interest in bi-parametric MRI, with T2-weighted images and DWI images only, leaving out DCE-MRI completely [34,35]. This would have advantages in terms of cost, efficiency, and accessibility [35]. However, the results of this study suggest that bi-parametric MRI would not produce optimal outcomes for African Americans and perhaps also be sub-optimal for other groups that are at risk for aggressive PCa. In addition, there is also increasing interest in quantitative MR imaging and analysis methods and artificial intelligence (AI) tools for PCa diagnosis [36,37]. Machine learning and other AI technologies can only perform well when constructed from representative and appropriate training sets. As such, training sets that do not include relevant information on potential confounding variables such as race may produce biased models. Therefore, bi-parametric MRI, quantitative MRI, and AI tools should be considered substitutes for conventional mpMRI and should account for the racial differences in prostate and PCa physiology and biology. Future studies should compare the current results with results for races or genetic sub-groups with lower incidences of PCa, e.g., the Asian population. MRI-based quantitative biomarkers that account for racial differences could be used to optimize the MRI protocols specifically for different racial groups. This will be a critical step towards personalized PCa screening. 

There are a few limitations to this study. First, this study is a retrospective single-center study with a relatively small sample size. Therefore, these results should be validated in a multi-center setting with a large sample size. Second, we used equipment and software from a single MR vendor. Similar studies are needed using scanners from other vendors as hardware and imaging parameters have been shown to affect quantitative MRI results [19,20,21]. Third, quantitative histology was performed on a subset of patients undergoing prostatectomy. Therefore, due to the small sample size, a comparison of quantitative histology in the two populations stratified by the Gleason score could not be performed. Fourth, we did not have the histological specimens to confirm an increased vessel density in cancers in African Americans, where a DCE analysis suggests increased blood flow [38]. This should be tested in future studies. Fifth, the use of a combined cohort of prostatectomy and biopsy can introduce potential bias while comparing the two populations when stratifying cancers by the Gleason score. This is due to the fact that a significant number of cancers are upgraded when biopsy subjects undergo prostatectomy [39].

## 5. Conclusions

This study shows that AAs have different quantitative DCE-MRI values for benign prostate and prostate cancer compared to CAs. Quantitative DCE-MRI is highly effective and improves PCa diagnosis in African Americans. These findings should be confirmed in larger studies, but we believe they are particularly important to consider, especially as bi-parametric MRI, which excludes DCE, is attracting increasing interest as a replacement for mpMRI. There are also some quantitative histologic differences between the cancers in AAs and CAs. The results also suggest that other racial groups may have differing prostate biology and physiology and may require specialized MRI screening methods. Racial differences should be taken into account when creating screening or diagnostic guidelines, particularly as bi-parametric MRI is being increasingly suggested for population screening.

## Figures and Tables

**Figure 1 cancers-16-03499-f001:**
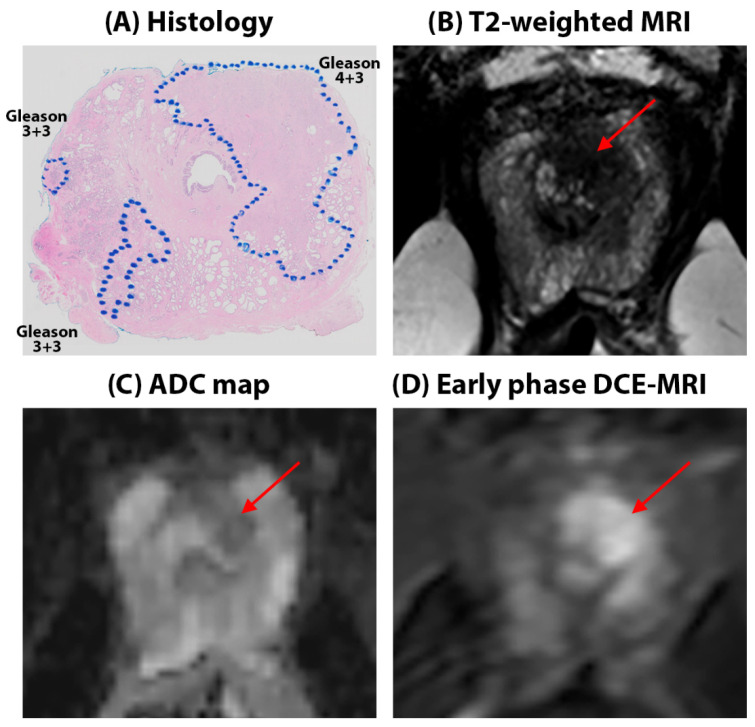
52-year-old African American patient with Gleason 4 + 3 cancer in the left apex in the peripheral zone (red arrows on MRI). The lesion is seen as a hypo-intense region on the T2W image, T2 (87.9 ± 16.4 ms), and mildly hypo-intense on ADC (1.32 ± 0.20 µm^2^/ms) maps with early focal enhancement on DCE-MRI, evidenced by high signal enhancement rate (19.3 s^−1^) and rapid washout rate (0.07 s^−1^). Surrounding benign tissue in the peripheral zone had ADC = 2.05 ± 0.10 µm^2^/ms, T2 = 308.9 ± 62.6 ms, α = 2.87 s^−1^, and β = 0.04 s^−1^. Another relevant finding is the presence of Gleason 3 + 3 cancers in the right apex. Cancers are outlined in blue on histology sections.

**Figure 2 cancers-16-03499-f002:**
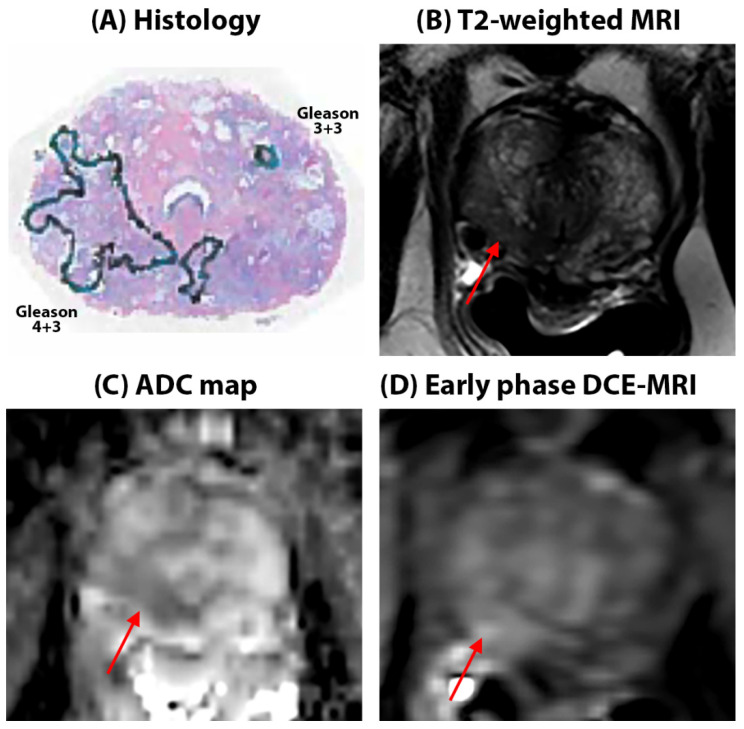
52-year-old Caucasian American patient with Gleason 3 + 4 cancer in the right apex in the peripheral zone (red arrows on mpMRI). The lesion is seen as a hypo-intense region on the T2W image, T2 (112.4 ± 54.6 ms), and mildly hypo-intense on ADC (0.86 ± 0.12 µm^2^/ms) maps with only diffuse early enhancement on DCE-MRI, evidenced by low signal enhancement rate (3.50 s^−1^) and washout rate (0.03 s^−1^). Surrounding benign tissue in the peripheral zone had ADC = 1.16 ± 0.19 µm^2^/ms, T2 = 125.1 ± 34.5 ms, α = 3.20 s^−1^, and β = 0.01 s^−1^. Cancers are outlined in blue on histology sections.

**Table 1 cancers-16-03499-t001:** Typical MR imaging parameters.

Imaging Sequence	Pulse Sequence	FOV (mm)	Scan Matrix Size	In-Plane Resolution (mm)	TE (ms)	TR (ms)	Slice Thickness (mm)	Flip Angle (°)
**Axial T2W**	SE-TSE	180 × 180	450 × 450	0.4 × 0.4	115 or 150	4800	3	90
**Multi-echo T2W (T2 mapping)**	SE-TSE	160 × 160	212 × 212	0.75 × 0.75	30, 60, 90, 120, 150, 180, 210, 240, 270	7850	3	90
**DWI ^a^**	SE-EPI	180 × 180	120 × 120	1.5 × 1.5	80	5000	3	90
**DCE-MRI ^b^**	T1-FFE	220 × 260	148 × 171	1.5 × 1.5	1.5	3.1	3	10

SE—spin echo; TSE—turbo spin echo; EPI—echo planar imaging; FFE—fast field echo; DCE-MRI—dynamic contrast-enhanced MRI. ^a^ b-values used: 0, 50, 150, 990, and 1500 s/mm^2^. ^b^ Contrast agent: gadoterate meglumine (Dotarem, Guerbet, France) or gadobenate dimeglumine (Multihance, Bracco, Minneapolis, MN, USA) was injected at a rate of 2.0 mL/s followed by a 20 mL saline flush. The contrast dose amount was based on the patient’s weight (0.1 mmol/kg). DCE-MRI T1-weighted images were taken with a temporal resolution of ~6.4 s at 60 dynamic scan points over 6.4 min.

**Table 2 cancers-16-03499-t002:** Patient characteristics.

		African Americans	Caucasian Americans	*p*
Age (years)		60 ± 7	58 ± 8	0.29
PSA (ng/mL)		8.5 ± 5.9	8.6 ± 9.9	0.94
Patients	Biopsy	25 (53%)	51 (52%)	0.89
	Prostatectomy	22 (47%)	47 (48%)	
	Total	47	98	-
ROIs	Cancer	45 (32%)	99 (34%)	0.81
	Benign	94 (68%)	196 (66%)	
	Total	139	295	-
ISUP grade group (Gleason score)	1 (Gleason 3 + 3)	13 (29%)	29 (29%)	0.01
	2 (Gleason 3 + 4)	16 (36%)	56 (57%)	
	3 (Gleason 4 + 3)	10 (22%)	11 (11%)	
	4 (Gleason 4 + 4)	5 (11%)	1 (1%)	
	5 (Gleason 4 + 5)	1 (2%)	2 (2%)	

**Table 3 cancers-16-03499-t003:** Quantitative mpMRI results from both populations.

		African Americans	Caucasian Americans	*p*-Value
ADC (µm^2^/ms)	Benign	1.53 ± 0.37	1.62 ± 0.37	0.12
	Cancer	1.03 ± 0.32	1.07 ± 0.34	0.75
	Gleason 3 + 3	1.23 ± 0.27	1.22 ± 0.34	0.88
	Gleason 3 + 4	1.12 ± 0.29	1.06 ± 0.33	0.55
	Gleason ≥ 4 + 3	0.85 ± 0.28	0.81 ± 0.27	0.62
T2 (ms)	Benign	151.9 ± 96.5	159.1 ± 73.2	0.70
	Cancer	107.5 ± 55.8	99.8 ± 27.5	0.25
	Gleason 3 + 3	110.8 ± 40.2	112.7 ± 31.2	0.89
	Gleason 3 + 4	108.0 ± 26.5	96.9 ± 26.6	0.31
	Gleason ≥ 4 + 3	104.9 ± 74.5	85.7 ± 12.8	0.35
DCE Signal enhancement amplitude or A (%)	**Benign**	**157.6 ± 82.6**	**126.1 ± 51.2**	**0.01**
	**Cancer**	**181.1 ± 44.8**	**120.2 ± 47.4**	**<0.001**
	**Gleason 3 + 3**	**200.9 ± 64.1**	**114.8 ± 28.44**	**<0.001**
	Gleason 3 + 4	158.5 ± 28.2	123.3 ± 55.6	0.18
	**Gleason ≥ 4 + 3**	**179.1 ± 35.3**	**125.4 ± 35.6**	**0.03**
DCE Signal enhancement rate (s^−1^)	Benign	5.1 ± 4.6	4.9 ± 2.9	0.81
	**Cancer**	**13.3 ± 9.3**	**6.1 ± 4.7**	**<0.001**
	**Gleason 3 + 3**	**10.3 ± 4.4**	**4.9 ± 2.9**	**0.002**
	**Gleason 3 + 4**	**11.7 ± 6.5**	**6.7 ± 5.6**	**0.04**
	**Gleason ≥ 4 + 3**	**15.6 ± 11.7**	**6.2 ± 2.9**	**0.02**
DCE Signal washout rate (s^−1^)	**Benign**	**0.01 ± 0.09**	**0.07 ± 0.07**	**<0.001**
	**Cancer**	**0.12 ± 0.07**	**0.07 ± 0.08**	**0.02**
	Gleason 3 + 3	0.10 ± 0.09	0.04 ± 0.08	0.12
	Gleason 3 + 4	0.11 ± 0.07	0.09 ± 0.08	0.62
	Gleason ≥ 4 + 3	0.13 ± 0.07	0.08 ± 0.03	0.13

**Table 4 cancers-16-03499-t004:** ROC analysis for differentiating cancer from benign prostate tissue.

	African Americans	Caucasian Americans	*p*-Value ^+^
ADC	AUC = 0.79 95% CI = [0.69, 0.88] Cutoff = 1.00 µm^2^/ms Sensitivity = 93% Specificity = 56%	AUC = 0.87 95% CI = [0.81, 0.92] Cutoff = 1.23 µm^2^/ms Sensitivity = 89% Specificity = 74%	0.06 *
T2	AUC = 0.68 95% CI = [0.56, 0.78] Cutoff = 136.2 ms Sensitivity = 45% Specificity = 88%	AUC = 0.79 95% CI = [0.72, 0.86] Cutoff = 116.9 ms Sensitivity = 67% Specificity = 80%	0.10 *
DCE Signal enhancement rate	AUC = 0.88 95% CI = [0.81, 0.96] Cutoff = 6.0 s^−1^ Sensitivity = 96% Specificity = 73%	AUC = 0.58* 95% CI = [0.48, 0.69] Cutoff = 6.3 s^−1^ Sensitivity = 38% Specificity = 79%	<0.001
DCE Signal washout rate	AUC = 0.81 95% CI = [0.73, 0.91] Cutoff = 0.02 s^−1^ Sensitivity = 96% Specificity = 52%	AUC = 0.50* 95% CI = [0.39, 0.60] Cutoff = 0.10 s^−1^ Sensitivity = 38% Specificity = 69%	<0.001

* not significant (*p* > 0.05). ^+^ *p*-value comparing if performance is better in one population over the other.

**Table 5 cancers-16-03499-t005:** Quantitative histology results from both populations.

		African Americans	Caucasian Americans	*p*-Value
Stroma	Benign	42.3 ± 10.2	43.1 ± 12.1	0.44
	Cancer	39.1 ± 11.5	38.6 ± 12.4	0.46
Epithelium	Benign	28.7 ± 9.0	29.6 ± 9.2	0.41
	**Cancer**	**44.7 ± 12.8**	**50.9 ± 12.3**	**0.04**
Lumen	Benign	28.8 ± 13.3	27.4 ± 11.1	0.38
	**Cancer**	**16.2 ± 6.8**	**10.5 ± 6.9**	**0.04**

## Data Availability

In accordance with the institutional review board, the data acquired in this study contain person-sensitive information, which can be shared only in the context of scientific collaborations.

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
