# Peer review of "Quantitative Multi-Parametric MRI of the Prostate Reveals Racial Differences"

_cancers, 2024, doi:10.3390/cancers16203499_

Round 1
Reviewer 1 Report
Comments and Suggestions for Authors
Well written manuscript with important findings regarding the role of DCE in mp MRI.
Study limited by small n, and limited RP samples to evaluate.
See comments and questions in attached .pdf proof

Author Response
Quantitative multi-parametric MRI of the prostate reveals racial differences
Response to reviewer comments
We really appreciate the comments from all the reviewers and editors in making this paper better for our audience. We hope you will find these revisions acceptable.
Reviewer 1
what about Asian Americans and Latin Americans in this study?
Again- perhaps some mention of other races, as opposed to a pure black/white dichotomy for PCa
Response: Thank you for mentioning this. We chose the 2 biggest population in our study. The incidence is slightly lower in Asian Americans and Latino Americans.
“Race is an important factor in prostate cancer risk and incidence. African Americans (AA) have higher risk for prostate cancer compared to Caucasian Americans (CA), while prostate cancer occurs less often in Asian American, Hispanic, and Latino men than in non-Hispanic White men.”
Therefore it wasn’t surprising to see less of Asians and Latinos, and a small sample size for this group led us to exclude them from our analysis.
.“We chose the two races in this cohort with the largest sample size: Caucasian Americans and African Americans for further analysis.”
Please provide % of each here for comparison # bx/n and #RP/n for both groups
Response: We have added this to the text and table 2.
Consider providing ISUP - Gleason Grade Groups for each
Response: We have added this to the table 2.
Please note in Table 5 that these are %'s . Q.- how do combinations of % stroma, epithelium, lumen add to 100% for benign and malignant tissue ? Is there another component not listed? How much of this tissue was malignant vs benign and how much of each tissue type was present in each? This is not clear from either the text under "Quantitative Histology", nor in Table 5.
Response: We are not calculating the percentage of the tissue that is malignant or cancer. Rather for cancer and benign ROIs we are segmenting them into stroma, epithelium and lumen: there 3 constituents tissue components that make up the prostate tissue. This is mentioned in the lines 162-167. We added more text to clarify this.
These tissue volumes all add up to a 100%, for eg – cancers in AA have 39.1 % stroma, 44.7 % epithelium and 16.2% lumen which adds up to 100%.
“ROIs were selected using ImageJ (National Institutes of Health, Bethesda, MD, USA) for confirmed cancers and benign tissue in the peripheral and transition zones corresponding to the ROIs taken for quantitative MRI analysis earlier. Using the “Smart Segment” functionality in Image Pro (Media Cybernetics, Rockville, Maryland), the images were segmented into the three constituents that make up prostate tissue: stroma, epithelium and lumen on the basis of color, intensity, morphology, and background.”
Doesn't lower lumen % in Caucasians suggest less differentiation to the authors? Yet there is not higher % of GGG 4 and 5 in Caucasians. Is there a theory for this possible discrepancy?
Response: There are 2 things at play – the GGG (amount of differentiation) and how dense the cancer is. Quantitative histology results where we found a lower percentage of lumen in CAs actually suggests more deviation form benign tissue as cancerous epithelium is replacing lumen. This suggests these cancer in CAs tend to be dense cancers. Despite the denser cancer in CA, they have less blood flow, suggesting less aggressive cancer. In addition, they tend to have lower GGG cancers, and thus we expect outcomes to be not as bad as AAs.
Gleason scoring or GGG depend on the tissue microarchitecture that quantitative histology cannot measure/account for.
Do the pathologists have an explanation for the finding of lower T2 and ADC in benign tissue in AA? Is it due to differential cellularity and blood supply of the BPH nodules? Or more BPH vs normal stroma in AA's?
Response: The T2 and ADC values were not significantly different in benign tissue for AA and CA. This could just be due to differences in ROI selection. If there are intrinsic differences (diffusivity and T2 of these components), that needs to be investigated further. We didn’t look at BPH, and stromal volume was similar in AA and CA.
BPH is commonly associated with acute and chronic inflammation, as high as 98% in early reference Drach, G et al; would this not also affect both vascularity and cellularity in this tissue, and therefore affect appearance on several MRI parameters, esp contrast enhancement, diffusion, etc?
Response: We did not look at BPH specifically between the two races. We noted in our text inflammation affects MRI.
“From MRI literature we also know that inflammation affects T2 relaxometry and diffusion measures, and can mimic prostate cancer, making cancer diagnosis more difficult [28].”
This prior finding could be discussed more fully in the present study, in light of your histopathologic correlation to the imaging.
Response: We discussed this in lines 296-303.
“The observed difference in contrast uptake and washout can be attributed to tumor microenvironment. Most importantly, neo-angiogenesis or higher microvessel density (MVD) in cancer of AA subjects compared to CA has been reported [32]. These blood vessels produced by cancers allow increased blood flow which results in rapid contrast uptake and quick washout. The increased blood flow brings increased oxygen and nutrients that support tumor growth, invasion, and metastatic progression. This is consistent with findings of increased tumor progression and worse clinical outcomes in AAs [33].”
This is a very important finding and potential recommendation to the PIRADS guidelines, as clearly there has been a deemphasis of DCE in updated versions. Perhaps calling for a larger multi-center review in the conclusions is warranted, as there have been recent papers published citing the cost and time savings of bi-parametric vs mpMRI. as the authors have cited (Ref 34, 35 and others) emphasizing the potential loss of important diagnostic information by not including full suite of available parameters for future quantitative techniques is major highlight of this study.
small n tends to lower the statistical power, yet statistical differences were identified, suggesting the findings could be greater with a larger cohort.
Response: Thank you seeing the merits of this work.
Can additional cases undergo the histologic quantification analysis to improve the breadth of this study?
Response: We do not have the samples to do additional histologic quantification at this time. We would need prostatectomy specimens. We have noted this in the limitations (lines 336).
This is quite true, and would be better to compare biopsy results in both cohorts separate from RP specimens in both cohorts. Again some mention of expected results in other races would be of interest.
Response: We mentioned the other races in different places in this manuscript. However, we have no literature to speculate what results may be for other races. As such we refrained for adding additional comments.
Reviewer 2 Report
Comments and Suggestions for Authors
Quantitative multi-parametric MRI of the prostate reveals racial differences
The authors present a well-written, scientifically sound manuscript covering an area of high importance. I have no major comments, and only a small number of minor comments, which the authors may wish to consider. In my opinion, the work is of a high quality and subject to the following minor issues is worthy of acceptance.
COMMENTS
The Abstract provides a good summary of what follows, but I did need to quickly Google ADC, T2 and DCE. Ideally the Abstract should be self-contained. Appreciate that word count will be constrained but if there is any way to explain these terms, that would be ideal.
Table 2, is it possible to add the p-values from the text to this table?
MRI diagnostics are not my field, but I was fascinated to read that "inflammation affects T2 relaxometry and diffusion measures, and can mimic prostate cancer". This is an ongoing issue for other methods for diagnostic and prognostic measures, for example:
PSA, PMID: 35125113, which is sensitive but poorly specific, or 'omics biomarkers, PMID: 38076065, which suffer the identical problem of being sensitive but poorly specific in the presence of confounding disorders such as BPH. This may not need to be mentioned in the context of this manuscript, but I thought it was interesting that the confounding effects of inflammation are a shared problem for all diagnostic and prognostic methods for PCA (for MRI and also for proteomics, e.g. over-expression of complement proteins). This is a strong argument for tailoring thresholds and levels to sub-populations that have different profiles.
Line 316, the authors briefly comment on AI, I would strengthen this to state that Machine Learning and other AI technologies can only perform well when constructed from representative and appropriate training sets. Related to the point above, training sets that do not include relevant information on potential confounding variables such as CA and AA will produce biased models.
Finally, there are limitations around the research, especially with regard to statistical power and possible bias from the fact that AAs are less likely to receive prompt diagnosis (and therefore be included in such a study at the same stage of disease as CAs), but I cannot see a way to avoid this in a preliminary study, the authors do an excellent job of noting the limitations, and this is where follow-up work is needed.
In conclusion, I very much enjoyed reading this manuscript and believe that the material should be published.
Author Response
Quantitative multi-parametric MRI of the prostate reveals racial differences
Response to reviewer comments
We really appreciate the comments from all the reviewers and editors in making this paper better for our audience. We hope you will find these revisions acceptable.
Reviewer 2
The authors present a well-written, scientifically sound manuscript covering an area of high importance. I have no major comments, and only a small number of minor comments, which the authors may wish to consider. In my opinion, the work is of a high quality and subject to the following minor issues is worthy of acceptance.
COMMENTS
The Abstract provides a good summary of what follows, but I did need to quickly Google ADC, T2 and DCE. Ideally the Abstract should be self-contained. Appreciate that word count will be constrained but if there is any way to explain these terms, that would be ideal.
Response: Thank you for pointing this out. We have expanded the abbreviations. We also added more information about quantitative T2 values in line 147/
Table 2, is it possible to add the p-values from the text to this table?
Response: We have added the p-values.
MRI diagnostics are not my field, but I was fascinated to read that "inflammation affects T2 relaxometry and diffusion measures, and can mimic prostate cancer". This is an ongoing issue for other methods for diagnostic and prognostic measures, for example: PSA, PMID: 35125113, which is sensitive but poorly specific, or 'omics biomarkers, PMID: 38076065, which suffer the identical problem of being sensitive but poorly specific in the presence of confounding disorders such as BPH. This may not need to be mentioned in the context of this manuscript, but I thought it was interesting that the confounding effects of inflammation are a shared problem for all diagnostic and prognostic methods for PCA (for MRI and also for proteomics, e.g. over-expression of complement proteins). This is a strong argument for tailoring thresholds and levels to sub-populations that have different profiles.
Response: We really appreciate you for bringing this up. This is definitely a strong argument for tailoring thresholds and levels to sub-populations that have different profiles as you mentioned. However, we didn’t look at BPH specifically, as the intention was to look at benign tissue (which does includes BPH as well) and cancer which of clinical importance. Therefore, we refrained from making an conclusion or suggestions based of these results.
Line 316, the authors briefly comment on AI, I would strengthen this to state that Machine Learning and other AI technologies can only perform well when constructed from representative and appropriate training sets. Related to the point above, training sets that do not include relevant information on potential confounding variables such as CA and AA will produce biased models.
Response: This is a great suggestion. We have added this in the discussion.
Finally, there are limitations around the research, especially with regard to statistical power and possible bias from the fact that AAs are less likely to receive prompt diagnosis (and therefore be included in such a study at the same stage of disease as CAs), but I cannot see a way to avoid this in a preliminary study, the authors do an excellent job of noting the limitations, and this is where follow-up work is needed.
Response: We agree with this comment and as the review notes this limitation we have acknowledged that future follow up work needs to be done.
In conclusion, I very much enjoyed reading this manuscript and believe that the material should be published.
Response: Thank you for seeing merit of our work.